# Constructing Sparse Neural Architecture with Deterministic Ramanujan Graphs

## Abstract

We present a sparsely connected neural network architecture constructed using the theory of Ramanujan graphs which provide comparable performance to a dense network. The method can be considered as a before-training, deterministic, weight free, pruning at initialization (PaI) technique. The deterministic Ramanujan graphs occur either as Cayley graphs of certain algebraic groups or as Ramanujan $r$-coverings of the full $(k, l)$ bi-regular bipartite graph on $k + l$ vertices. Sparse networks are constructed for bipartite graphs representing both the convolution and the fully connected layers. We experimentally show that the proposed sparse architecture provides comparable accuracy with a lower sparsity ratio than those achieved by previous approaches based on non-deterministic methods for benchmark datasets. In addition, they retain other desirable properties such as path connectivity and symmetricity.

## 1 Introduction

Sparsely connected neural architectures are becoming increasingly popular for their reduced training time and comparable accuracy with a dense network. Existence of sparse high performing subnetworks of a backbone dense network forms the basis of the well known lottery ticket hypothesis (Frankle & Carbin, 2019). Several approaches have been directed towards identifying winning lottery tickets with a minimal effort. Initial research were based on applying established pruning algorithms on a partially trained network (Renda et al., 2020; Fischer & Burkholz, 2022). In order to reduce the effort even further, Pruning at Initialization (PaI) was suggested (Frankle et al., 2020; Wang et al., 2021; Sreenivasan et al., 2022). These method use the structure of the initialized network, in a data dependent or independent manner, to prune the network to a high sparsity ratio (Sreenivasan et al., 2022; Lee et al., 2019a;b; Wang et al., 2020; Tanaka et al., 2020).

Expander graphs are connected sparse networks (Hoory et al., 2006) with bounded expansion factors. Higher spectral gap between the first and the second eigenvalues of a graph adjacency matrix points towards a better expansion. Ramanujan graphs (Lubotzky et al., 1988) are a class of regular spectral expanders with maximally high spectral gaps. It has been empirically shown that the expansion property is strongly correlated with the performance of sparse neural networks (Prabhu et al., 2018b; Pal et al., 2022).

Spectral sparsification is a method of obtaining such expander like neural networks (Laenen, 2023). In general, the expander networks provide a sparse initialization architecture which may be trained to a high accuracy (Stewart et al., 2023; Esguerra et al., 2023; Prabhu et al., 2018b). Most of the expander networks used for this purpose are obtained by first generating random bipartite graphs for each layer, and then selecting the ones with a large spectral gap. This is based on the fact that random graphs are weakly Ramanujan (a conjecture of Alon, proved by Freidman). However, the expander based techniques mentioned above often favors random network initialization which are sensitive to random reinitialization and rewiring (Ma et al., 2021). Additional spectral measures are necessary to arrest these possibilities (Hoang et al., 2023). The iterative mean difference of bound (IMDB) score proposed for this purpose in (Hoang et al., 2023) is an elegant measure which correlates with the performance of a sparse networks.

We propose a deterministic sparse network initialization technique based on Ramanujan graphs that are constructed either as Cayley graphs of certain algebraic groups or as Ramanujan $r$-coverings of the full $(k, l)$ bi-regular bipartite graph on $k + l$ vertices. Prior approaches to using Ramanujan

expander graphs for PaI have predominantly relied on constructions based on random or iterated magnitude pruning techniques. Unfortunately, this approach has led to the formation of irregular graph networks that do not strictly adhere to the rigorous definition of Ramanujan graphs. Our approach of constructing a deterministic Ramanujan network avoids the irregularity issues and thus naturally have a high IMDB value. Ramanujan initializers using these bipartite graphs suitably represent the fully connected as well as the convolutional layers.

Deterministic Ramanujan graph based sparse network initialization has several advantages. Path connectedness and regularity is guaranteed by the Ramanujan graph construction technique. This ensures good performance even at very low remaining weight ratios. The sparse networks generated are data independent, structurally pre-defined, with a static mask across the training iterations. A symmetry property of the sparse adjacency matrix is preserved in this process.

Experimental results on benchmark image classification data sets show that Ramanujan sparse network initialization provides comparable performance with dense networks. The paper is organized as follows. We present a brief literature survey in the next section. Contributions of the paper are highlighted next. The mathematical formulation of deterministic Ramanujan graphs is then presented, along with the construction techniques of sparse neural network layers. Finally, the experimental results are outlined.

## 2 RELATED WORK

Pruning at initialization (PaI) has been well studied in literature both in data dependent and data independent contexts (Cheng et al., 2023). The baseline consists of random pruning techniques based on either uniform edge sampling or Erdos-Renyi graphs (Liu et al., 2022; Evci et al., 2020; Mocanu et al., 2018; Gadhikar et al., 2023). More advanced techniques like SNIP use edge sensitivities (Lee et al., 2019b). Gradient flows over the edge weights are used in state-of-art techniques like GraSP Wang et al. (2020), and SynFlow (Tanaka et al., 2020). Other gradient sensitivity scores has also been used for this purpose (Ramanujan et al., 2020).

Expander based random winning lottery ticket generation has been studied in (Stewart et al., 2023). The methodology is based on generating random $d$-regular graphs for the bipartite layers. These graphs are Ramanujan with a high probability. A deep expander sparse network, the X-Net, is presented in (Prabhu et al., 2018b). It is constructed by sampling $d$-left regular graphs from the space of all bipartite graphs. Ramanujan graph based sparsity aware network initialization is proposed in (Esguerra et al., 2023). It uses an orthogonalization technique for block sparse bi-adjacency matrix construction of the fully connected as well as convolutional layers. The method preserves path connectedness and dynamical isometry of gradient descent.

One-shot neural network pruning using spectral sparsification is presented in (Laenen, 2023). It is based on the effective resistance algorithm for obtaining spectrally sparse bipatite graphs. It is shown that this is equivalent to global sparsification. RadiX-Net (Kepner & Robinett, 2019) is a deterministic sparse neural architecture with mixed-radix topologies. It has desirable symmetry properties that preserves path connectedness and eliminates training bias. Connectivity properties of networks are used in other graph theoretic initialization schemes that define an initial sparse network topology (Vysogorets & Kempe, 2023; Chen et al., 2022; 2023).

## 3 RESEARCH GAP AND CONTRIBUTIONS

Recently, there has been a flurry of works on the construction of pruned sparse networks based on various graph theoretic properties including expansion, path-connectivity, symmetry etc. (Prabhu et al., 2018a; Kepner & Robinett, 2019; Pal et al., 2022; Stewart et al., 2023; Hoang et al., 2023). However, none of the above works could guarantee the following three properties at the same time: (i) Ramanujan property - allows us to construct the best possible expanders given a set of vertices and maintaining a high level of sparsity, (ii) Path-connectedness - a desirable property for all PaI architectures, and, (iii) Highly symmetric - a desirable property for computational purposes.

Further, all the previous approaches based on random network initializations or existent pruning strategies suffer from the issue of irregularity and are not guaranteed to be rigorously Ramanujan. For instance, application of the work of Hoory (Hoory et al., 2006) as mentioned in (Hoang et al.,

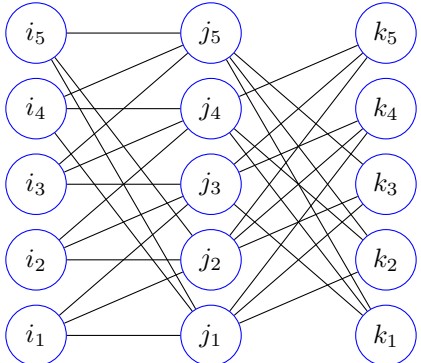

Figure 1: An example of a double layered regular bipartite graph where each layer is Ramanujan.

2023; Pal et al., 2022) etc depends on the crucial fact that the minimal degrees of the base bipartite graphs needs to be $\geq 2$ for the graphs to be called Ramanujan. Our architecture based on deterministic regular Ramanujan graphs of degree $\geq 3$ ensures that the initialized networks remain Ramanujan, are path-connected and are highly symmetric being either Cayley graphs of certain algebraic groups to replace the balanced dense bipartite graphs or the Ramanujan $r$-covering of full bi-regular bipartite graphs to replace the unbalanced dense bipartite graphs. *This is the first such implementation of deterministic Ramanujan graph neural networks.*

Path-connectedness: The fact that each layer of the bipartite graphs are either regular or bi-regular with the regularity bigger than $3$ ensures that the entire architecture remains path-connected, i.e., starting from any node in the first layer we can reach a node in the last layer by a connected path. A proof of this is direct. In Figure 1, suppose there are 3 layers $I, J, K$. We wish to reach layer $K$ starting from any point in layer $I$ by a connected path. Pick any $i_r, r \in \{1, 2, 3, 4, 5\}$. Use the fact that there is at least one edge going out from $i_r$ to reach some $j_s$ and from $j_s$ again use the same fact of outgoing edges bigger than 1 to reach some point in layer $K$. The general case follows by induction on the number of layers.

High-symmetricity: The adjacency matrices of Cayley graphs and that of covers of Cayley graphs have much more symmetry than that of general regular graphs. Often computations are optimised to use such symmetry viz. in the case of the software GAP (Group, 2022) for instance. This may lead to fast computation on sparse Ramanujan Cayley graphs. In the above graph, the adjacency matrix of the first layer (which can be represented as a Cayley graph on the group $\mathbf{Z}_2 \times \mathbf{Z}_5 = \{(x, y) : 0 \leq x \leq 1, 0 \leq y \leq 4\}$ with generating set $S = \{(1, 0), (1, 1), (1, 4)\}$) is $Adj = \begin{pmatrix} 0_{5 \times 5} & B_{5 \times 5} \\ B_{5 \times 5}^T & 0_{5 \times 5} \end{pmatrix}$

where $B = \begin{pmatrix} 1 & 1 & 1 & 0 & 0 \\ 0 & 1 & 1 & 1 & 0 \\ 0 & 0 & 1 & 1 & 1 \\ 1 & 0 & 0 & 1 & 1 \\ 1 & 1 & 0 & 0 & 1 \end{pmatrix}$.

The principal contributions of the paper are:

1. We present a deterministic Ramanujan graph construction technique for initializing sparse neural networks. To the best of our knowledge, no other work exists towards this direction.

2. The construction technique is adapted for both fully connected and convolution layers.

3. Experimental results demonstrate the effectiveness of deterministic Ramanujan graph initialized networks on benchmark datasets at very high sparsity.

## 4 FORMULATION OF SPARSE NEURAL RAMANUJAN GRAPHS

In this section we present the mathematical framework which allows us to construct in a deterministic manner the sparse sub-network of the original dense neural networks. This forms the basis of our

strategy of pruning at initialization. Recall that a Ramanujan graph is an extremal expander graph in the sense that its spectral gap is almost as large as possible. For this article, we shall be concerned with bipartite Ramanujan graphs.

**Definition 1 (Bipartite Ramanujan graphs)** *Let $\Gamma = (V, E)$ be a $d$-regular ($d \geq 3$) bipartite graph. Let the eigenvalues of its adjacency matrix be $\lambda_n \leq \lambda_{n-1} \leq \ldots \leq \lambda_2 \leq \lambda_1$. Then $\Gamma$ is said to be Ramanujan iff $|\lambda_i| \leq 2\sqrt{d-1}$, for $i = 2, \ldots, (n-1)$.*

Note that we are considering undirected graphs, so the adjacency matrix is a $0 - 1$ symmetric matrix and the eigenvalues are all real. A non-bipartite graph is said to be Ramanujan if in addition $|\lambda_n| \leq 2\sqrt{d-1}$. For an unbalanced $(d_1, d_2)-$biregular bipartite graph ($d_1, d_2 \geq 3$), the condition of being Ramanujan changes to $|\lambda_i| \leq \sqrt{d_1 - 1} + \sqrt{d_2 - 1}$, for $i = 2, \ldots, (n - 1)$. We see that when $d_1 = d_2$, it transforms to the usual definition. A detailed description of Ramanujan graphs can be found in (Hoory et al., 2006, sec. 5.3).

The above graphs are excellent spectral expanders. They are also notoriously difficult to construct. In fact, even the question of existence of (infinite families of) Ramanujan graphs is a non-trivial one and it is not yet fully resolved for the non-bipartite case. The first such construction of graphs are due to Lubotzky–Phillips–Sarnak (LPS) (Lubotzky et al., 1988) (and independently by Margulis (Margulis, 1988)). We shall modelise our initial pruned network according to these constructions.

### 4.1 MATHEMATICAL PRELIMINARIES

For the construction of the deterministic Ramanujan networks, we shall need the following notions from arithmetic.

**Definition 2 (Quadratic residue and Legendre symbol)** *An integer $q$ is called a quadratic residue modulo $n$ if there exists an integer $x$ such that $x^2 \equiv q \pmod{n}$. Otherwise, $q$ is called a quadratic non-residue modulo $n$.*
*Let $p$ be an odd prime number and $a$ be an integer. The Legendre symbol of $a$ and $p$ is defined as*

$$\left(\frac{a}{p}\right) = \begin{cases} 1 & \text{if } a \text{ is a quadratic residue modulo } p \text{ and } a \not\equiv 0 \pmod{p}, \\ -1 & \text{if } a \text{ is a quadratic non-residue modulo } p, \\ 0 & \text{if } a \equiv 0 \pmod{p}. \end{cases}$$

Given a prime $a$, there are infinitely many primes $p$ such that Legendre symbol of $a$ and $p$ is $-1$ (and also there are infinite many primes $p$ such that it is $+1$)

**Definition 3 ($PGL_2(K)$)** *Let $K$ be a field. Let us denote by $GL_2(K)$ the group of invertible 2-by-2 matrices with coefficients in $K$, ie, the matrices with non-zero determinant. Let $PGL_2(K)$ be the quotient group*

$$PGL_2(K) = GL_2(K)/Z(K)$$

*where*

$$GL_2(K) = \left\{ \begin{pmatrix} a & b \\ c & d \end{pmatrix} : ad - bc \neq 0 \,(in\ K) \right\}, Z(K) = \left\{ \begin{pmatrix} a & 0 \\ 0 & a \end{pmatrix} : a \neq 0 (in\ K) \right\}$$

Remark: If $K = \mathbb{F}_q$, then $|PGL_2(K)| = q(q^2 - 1)$. In Section 4.3, we shall use this property to construct bipartite $\frac{q(q^2-1)}{2}$ by $\frac{q(q^2-1)}{2}$ Ramanujan networks.

### 4.2 REGULAR RAMANUJAN GRAPHS

Let $p, q \equiv 1 (mod\ 4)$ be distinct odd primes (the condition of $1(mod\ 4)$ can be removed at the cost of making the analysis more technical and complicated, we shall mention later how it is achieved). The graph $X^{p,q}$ is constructed using the following general method.

1. It is a Cayley graph on the subgroup of 2 by 2 matrices, $PGL_2(\mathbb{F}_q)$ where $\mathbb{F}_q$ is the finite field of characteristic $q$.

2. Consider the equation $a_0^2 + a_1^2 + a_2^2 + a_3^2 = p$. Jacobi's four square theorem states that there are $p + 1$ solutions to the equation $a_0^2 + a_1^2 + a_2^2 + a_3^2 = p$ with $a_0 > 0$ odd (i.e., $a_0 \equiv 1 \pmod 2$) and $a_1, a_2, a_3$ even. Now, for each such solution $(a_0, a_1, a_2, a_3)$ consider the matrix $\begin{pmatrix} a_0 + ia_1 & a_2 + ia_3 \\ -a_2 + ia_3 & a_0 - ia_1 \end{pmatrix}$ where $i$ is some fixed solution to $i^2 = -1 \pmod q$. This matrix belongs to $PGL_2(\mathbb{F}_q)$. This can be checked from the definition of $PGL_2(\mathbb{F}_q)$.

3. Form the generating set $S$ of the Cayley graph to be the set of these $(p+1)$ matrices. Thus $X^{p,q} = Cay(PGL_2(\mathbb{F}_q), S)$.

4. The graphs are bipartite iff $p$ is not a quadratic residue modulo $q$ or in other words the Legendre symbol $\left(\frac{q}{p}\right) = -1$. The bipartite graphs $X^{p,q}$ will be $(p+1)$-regular, of size $\frac{q(q^2-1)}{2}$ by $\frac{q(q^2-1)}{2}$ and are Ramanujan (Lubotzky et al., 1988).

Remark: If $p \equiv 3 \pmod 4$, then a similar strategy is employed, except in this case one looks at solutions of $a_0^2 + a_1^2 + a_2^2 + a_3^2 = p$ with $a_0 \equiv 0 \pmod 2$ (Musitelli & de la Harpe, 2006, sec. 2).

### 4.3 CONSTRUCTION OF THE FULLY CONNECTED LAYERS

For the fully connected layers consisting of balanced bipartite graphs, we prune them at initialization in accordance with the Ramanujan graph structure of LPS. For this we select a prime $q$ such that $\frac{q(q^2-1)}{2}$ by $\frac{q(q^2-1)}{2}$ is closest to the size of the original bipartite layer. We then select the prime $p$ such that the Legendre symbol $\left(\frac{q}{p}\right) = -1$ (note that this choice is always possible as given a prime $q$ there are infinite number of primes $p$ satisfying this property). Selecting the minimum possible value of $p$ will give us the sparsest Ramanujan graph. For a 4096 by 4096 original network, our choice of $(p, q) = (5, 17)$ giving rise to 6 regular bipartite sparse Ramanujan networks. Note that here we have taken $p \equiv 1 \pmod 4$, but we could have also chosen $p = 3$ or even $p = 2$ (see construction of cubic Ramanujan graphs (Chiu, 1992)) resulting in even sparser networks.

### 4.4 BI-REGULAR RAMANUJAN GRAPHS

A bipartite graph is said to be $(d_1, d_2)$ bi-regular if each bi-partition has fixed regularity $d_1$, $d_2$ respectively. Note that a simple computation reveals that if $(n_1, n_2)$ are the bi-partition sizes, then $n_1 d_1 = n_2 d_2$. Thus three parameters are needed to specify these types of graphs. One way to construct bi-regular Ramanujan graphs is the following, see (Burnwal et al., 2021):
Fix a prime $q$ and a $q \times q$ cyclic shift permutation matrix $P = [P]_{ij}$ with $[P]_{ij} = 1$ if $j = i - 1 \pmod q$ and 0 otherwise. Recall that the adjacency matrix of any $m \times n$ bipartite graph can be written as $Adj = \begin{pmatrix} 0_{m \times m} & B_{m \times n} \\ B_{m \times n}^T & 0_{n \times n} \end{pmatrix}$, where $B$ is called the bi-adjacency matrix. Define the bi-adjacency matrix of this bipartite graph to be $B = \begin{pmatrix} I_q & I_q & \dots & I_q \\ I_q & P & \dots & P^{l-1} \\ I_q & P^2 & \dots & P^{2(l-1)} \\ \vdots & & & \\ I_q & P^{q-1} & \dots & P^{(q-1)(l-1)} \end{pmatrix}$ where $I_q$ is the $q \times q$ identity matrix and $P$ is as above. $B$ is a $q^2 \times lq$ matrix and the bipartite graph is either $q^2 \times lq$ with bi-regularity $(l, q)$ or symmetrically $lq \times q^2$ with bi-regularity $(q, l)$. The graphs whose bi-adjacency matrices are represented as $B$ (or $B^T$) are Ramanujan. These graphs are explicit realisations of the Ramanujan $r$-coverings of the full $(k, l)$ bi-regular bipartite graph on $k + l$ vertices as shown in (Hall et al., 2018, cor 2.2).

### 4.5 CONSTRUCTION OF THE CONVOLUTION LAYERS

For pruning the convolution layers, we utilise the bi-regular Ramanujan graphs. Let $q \geq l$. We analyse the size of the pruned network compared to the original fully connected network. The total number of edges in the $q^2 \times lq$ Ramanujan graph is $lq^2$ whereas the original network has $lq^3$ edges.

Choosing the value of $q$ to be as large as possible ensures that the pruned network has a small percentage of edges remaining while still being a Ramanujan network.

The condition $q \geq l$ is not a necessary requirement for implementing the technique outlined in Section 4.3. The reason behind this flexibility lies in the specific properties of the unbalanced bipartite graph $B$, which has dimensions $q^2$ by $lq$. The critical insight is that if the unbalanced bipartite graph $B$ is Ramanujan, then its transpose, denoted as $B^T$ (with dimensions $lq$ by $q^2$), is also Ramanujan.

A $m$ by $n$ bipartite graph connects two network layers with $m$ and $n$ nodes respectively. The technique for convolution layers to generate bi-regular bipartite graphs (of order $m$ by $n$) has complexity $O(mn)$. This complexity is due to the creation of the pruning mask which is of size $m \times n$. The LPS technique to generate $p + 1$ regular Ramanujan graphs has complexity $O(q^5 + p^4) = O(q^5) = O(m^{5/3})$. Here $m = n = O(q^3)$. This complexity is due to the creation of the $PGL_2$ group in which first we need to create the generator matrix and then find the equivalence classes which takes time $O(q^4q)$. The solution to the four square problem has complexity $O(p^4)$. Since the number of nodes are much less compared to the total parameters, the complexity is low.

### 4.6  General bipartite networks

In the case of bipartite networks with arbitrary sizes, one can achieve as sparse Ramanujan graphs as possible. It has recently been proven by Marcus, Spielman and Srivastava that for the regular case, for each degree $d \geq 3$, infinite families of bipartite Ramanujan graphs exist. This is also true for the bi-regular case, for each pair $(d_1, d_2)$ with $d_1, d_2 \geq 3, d_2 = kd_1, k \geq 2$. Further they showed the existence of these types of graphs of all sizes. Their method of proof is probabilistic and existential in nature. It does not give explicit families of bipartite Ramanujan graphs. However there now exist polynomial time algorithms (Cohen, 2016) (for the regular case) (Gribinski & Marcus, 2021) (for the bi-regular case) with which we can extract explicit Ramanujan graphs. For the regular case, we fix an integer $n \geq 3$ and a degree $d \geq 3$ and in the output we shall obtain a $d$-regular $n \times n$ bipartite Ramanujan graph while for the bi-regular case, we fix three integers $n, k, d$ with $n > 2, d > 2, k \geq 2$ and obtain $(d, kd)$ bi-regular Ramanujan graph of size $kn \times n$.

## 5  Experimental Methodology and Results

The goal of our experiments is to study the effectiveness of deterministic Ramanujan graph based sparse network initialization. We compare the classification accuracy of the models at very low remaining weight percentages. It is desirable that the sparse networks have a small accuracy drop ($\delta_{acc}$) as compared to the dense networks. We now present details of our experimental studies.

### 5.1  Datasets and Architectures

The datasets used for the experiments are Cifar-10 and Cifar-100 (Krizhevsky, 2009). The experiments are performed over a variety of architectures including VGG13, VGG16, VGG19 (Simonyan & Zisserman, 2014), AlexNet (Krizhevsky et al., 2012), ResNet18 and ResNet34 (He et al., 2016) to show the robustness of our PaI method. We proceed in two parts. In the first part we replace the intermediate Fully Connected Layer by sparse Ramanujan layers. This is applicable to the following architectures VGG13, VGG16, VGG19 and AlexNet, on both the Cifar-10 and Cifar-100 datasets. In the second part while pruning the whole network including the convolution layers, the experiments have been performed on Cifar-10 dataset for architectures VGG13, VGG16, VGG19, AlexNet, ResNet18, ResNet34 and on Cifar-100 dataset for architectures ResNet18 and ResNet34.

### 5.2  Experimental Methodology

The experiments are conducted in two parts: 1) Only pruning the Fully Connected Layer, 2) Pruning the whole network including the Convolution layers. For the first part, the construction of Ramanujan graphs as given in Section 4.3 is used. Only the VGG and AlexNet architectures have a dedicated fully connected layer in-between the network and the values used for $p$ and $q$ (Section 4.2) are given in Table 1. This results in the fully connected layer becoming of size $2448 \times 2448 = q \times (q^2 - 1)/2$ with the effective number of connections between the bipartite graph being equal to $2448 \times (p + 1)$.

Table 1: Values of $p$ and $q$ used to generate the sparse fully connected layer

| Model | VGG13, VGG16, VGG19, AlexNet |
|---|---|
| FC Layer Size | $4096 \times 4096$ |
| $p$ | 5, 29, 109 |
| $q$ | 17 |

For the second part of the experiment, the convolution layers are pruned according to the construction given in Section 4.5. The convolution layer can be thought of as a matrix of dimensions $|N_{out}| \times |N_{in}| \times |K_w| \times |K_h|$ where $|N_{out}|$ is the number of output channels, $|N_{in}|$ is the number of input channels, $|K_w|$ is the kernel width and $|K_h|$ is the kernel height. This is considered to be a bipartite graph with $|V_{left}| = |N_{in}| \times |K_w| \times |K_h|$ and $V_{right} = |N_{out}|$ where each vertex of $V_{left}$ has an edge with each vertex of $V_{right}$. The size of convolution layer being pruned and the choice of $l$ and $q$ for different architectures is given in Table 2.

Table 2: Values of $q$ and $l$ to generate Ramanujan graphs for layers of VGG, AlexNet and ResNet

| VGG13 | | | VGG16 | | | VGG19 | | |
|---|---|---|---|---|---|---|---|---|
| Conv Size | $q$ | $l$ | Conv Size | $q$ | $l$ | Conv Size | $q$ | $l$ |
| $[256 \times 256 \times 3 \times 3] \times 1$ | 13 | 177 | $[256 \times 256 \times 3 \times 3] \times 2$ | 13 | 177 | $[256 \times 256 \times 3 \times 3] \times 3$ | 13 | 177 |
| $[512 \times 256 \times 3 \times 3] \times 1$ | 19 | 121 | $[512 \times 256 \times 3 \times 3] \times 1$ | 19 | 121 | $[512 \times 256 \times 3 \times 3] \times 1$ | 19 | 121 |
| $[512 \times 512 \times 3 \times 3] \times 3$ | 19 | 242 | $[512 \times 512 \times 3 \times 3] \times 5$ | 19 | 242 | $[512 \times 512 \times 3 \times 3] \times 7$ | 19 | 242 |
| Conv to Linear Size | $q$ | $l$ | Conv to Linear Size | $q$ | $l$ | Conv to Linear Size | $q$ | $l$ |
| $2448 \times 25088$ | 47 | 533 | $2448 \times 25088$ | 47 | 533 | $2448 \times 25088$ | 47 | 533 |
| AlexNet | | | ResNet18 | | | ResNet34 | | |
| Conv Size | $q$ | $l$ | Conv Size | $q$ | $l$ | Conv Size | $q$ | $l$ |
| $[384 \times 256 \times 3 \times 3] \times 1$ | 19 | 121 | $[64 \times 64 \times 3 \times 3] \times 4$ | 7 | 82 | $[64 \times 64 \times 3 \times 3] \times 6$ | 7 | 82 |
| $[384 \times 384 \times 3 \times 3] \times 1$ | 19 | 181 | $[128 \times 64 \times 3 \times 3] \times 1$ | 11 | 52 | $[128 \times 64 \times 3 \times 3] \times 1$ | 11 | 52 |
| $[256 \times 384 \times 3 \times 3] \times 1$ | 13 | 265 | $[128 \times 128 \times 3 \times 3] \times 3$ | 11 | 104 | $[128 \times 128 \times 3 \times 3] \times 7$ | 11 | 104 |
| | | | $[256 \times 128 \times 3 \times 3] \times 1$ | 13 | 88 | $[256 \times 128 \times 3 \times 3] \times 1$ | 13 | 88 |
| | | | $[256 \times 256 \times 3 \times 3] \times 3$ | 13 | 177 | $[256 \times 256 \times 3 \times 3] \times 11$ | 13 | 177 |
| | | | $[512 \times 256 \times 3 \times 3] \times 1$ | 19 | 121 | $[512 \times 256 \times 3 \times 3] \times 1$ | 19 | 121 |
| | | | $[512 \times 512 \times 3 \times 3] \times 3$ | 19 | 242 | $[512 \times 512 \times 3 \times 3] \times 5$ | 19 | 242 |
| Conv to Linear Size | $q$ | $l$ | | | | | | |
| $2448 \times 25088$ | 47 | 533 | | | | | | |

The pruning mask thus obtained is a matrix of size $q^2 \times lq$ with the effective connections being $q^2 \times l$. The original pruning mask of the convolution layer has size $|N_{out}| \times [|N_in| \times |K_w| \times |K_h|]$. By construction the obtained Ramanujan graph is actually a subgraph of the original pruning mask and thus the entries in the original mask not part of the constructed Ramanujan graph are set to 0.

The network density reported in the tables of Section 5.4 is calculated by dividing the number of effective connections which is equal to the sum of $q \times (q^2 - 1)/2 \times (p + 1)$ (effective number of connections in fully connected layer) and $q^2 \times l$ (effective number of connections in each of the convolution layers) divided by the total number of connections present in the unpruned network.

Training parameters for all of the architectures are same and are summarized in Table 3. We train the networks for 30 epochs for our methods in contrast to 250 epochs used by (Hoang et al., 2023).

Table 3: Training parameters for the architectures compared

| Train Epochs | Batch Size | Learning Rate | Optimizer | Weight Decay | Momentum | Initialization |
|---|---|---|---|---|---|---|
| 30 | 64 | 0.003 | SGD | 0.005 | 0.9 | Kaiming Uniform |

## 5.3 METHODS COMPARED

The performance of the pruned networks are compared with that of similar dense networks. We have also compared our method against various Pruning at Initialization (PaI) techniques such as Random (Liu et al., 2022), ERK (Evci et al., 2020; Mocanu et al., 2018), SNIP (Lee et al., 2019b), GraSP (Wang et al., 2020), and SynFlow (Tanaka et al., 2020).

### 5.4 RESULTS AND DISCUSSION

We study the accuracy of the sparse networks obtained by our technique for various architectures and dataset at different levels of sparsification. The accuracy are compared with those of the corresponding unpruned network with similar number of nodes. The unpruned and the pruned networks are both trained for 30 epochs using the parameters reported in Table 3.

For the first part of the experiment where only the intermediate fully connected layer is pruned, the results are summarized in Table 4 for the Cifar-10 and Cifar-100 datasets. It can be observed that the Ramanujan graph construction allows us to extremely prune the fully connected layer upto $0.08\%$ while still retaining the accuracy as of the unpruned model. It can also be seen that for $1.6\%$ pruning the accuracy tends to actually beat the unpruned one suggesting that pruning also leads to a reduction in the overfitting of data for the smaller Cifar-10 dataset.

Table 4: Accuracy of VGG and AlexNet when only the intermediate fully connected layer is pruned

| Model | Cifar-10: FC Size / Network Density | | | |
|---|---|---|---|---|
| | $4096 \times 4096$ / Unpruned | $2448 \times 110$ / 1.6% | $2448 \times 30$ / 0.43% | $2448 \times 6$ / 0.08% |
| VGG13 | 83% | 83% | 82% | 83% |
| VGG16 | 84% | 86% | 84% | 83% |
| VGG19 | 85% | 86% | 84% | 83% |
| AlexNet | 81% | 82% | 81% | 81% |
| Model | Cifar-100: FC Size / Network Density | | | |
| | $4096 \times 4096$ / Unpruned | $2448 \times 110$ / 1.6% | $2448 \times 30$ / 0.43% | $2448 \times 6$ / 0.08% |
| VGG13 | 54% | 54% | 52% | 52% |
| VGG16 | 57% | 55% | 55% | 53% |
| VGG19 | 55% | 55% | 52% | 53% |
| AlexNet | 53% | 52% | 52% | 51% |

In the second part of the experiment wherein we prune the complete model including the convolution layers, we could achieve an overall pruning percentage of $1.7\%$ for VGG, $2.3\%$ for AlexNet and around $5\%$ for the ResNet architectures. The accuracy of the models on the Cifar-10 and Cifar-100 datasets are summarized in Table 5. It can be observed that pruned networks still maintain their accuracy with a slight reduction of around $2 - 3\%$ compared to the unpruned networks.

Table 5: Accuracy of various architectures when the complete network is pruned including the Convolution and the FC layers.

| Model | Unpruned accuracy | Accuracy / Network Density |
|---|---|---|
| | Cifar-10 dataset | |
| VGG13 | 83% | 81%/1.7% |
| VGG16 | 84% | 82%/1.7% |
| VGG19 | 85% | 83%/1.7% |
| AlexNet | 81% | 77%/2.3% |
| ResNet18 | 83% | 82%/5.6% |
| ResNet34 | 83% | 84%/5.2% |
| | Cifar-100 dataset | |
| ResNet18 | 56% | 54%/5.6% |
| ResNet34 | 57% | 56%/5.2% |

Next, we compare the performance of the proposed Ramanujan sparse network initializations with other state-of-art pruning at initialization techniques. The comparison of accuracy and drop in accuracy from the unpruned model ($\delta_{acc}$) at remaining weight percentages of $1.7\%$ and $5.2\%$ are shown in Table 6 for the VGG16 (Cifar-10) and ResNet34 (Cifar-10 and Cifar-100). The results for the related PaI methods, namely, Random pruning, ERK, SNIP, SynFlow and GraSP, have been reported from (Hoang et al., 2023). The accuracy of the unpruned network is also reported. Note that some of the base models of the architectures considered in their paper are different. For instance, in the case of VGG16, at least **two different models are coined by the same term in the literature**. **The original VGG16 architecture has** $135$**M parameters** and the accuracy of the trained original VGG network is around $84\%$ in the unpruned case. This is the network with which the accuracy of

our pruned network is compared. There is at least one other VGG16 network sometimes referred to by some authors as CIFAR-VGG, see for instance Giuste & Vizcarra, 2020 which is a modified (pretrained) VGG16 architecture with additional dropout and weight decay to reduce overfitting potential. These changes lead to higher classification accuracy of $94\%$ on the CIFAR-10 testing dataset, on which the results of (Hoang et al., 2023) were reported. We use the term VGG-base for the 135M parameter model. This choice was made to emphasize our ability to implement the method without additional information on dropout and pretraining, which the smaller VGG (15M parameters) utilizes. Since the original starting networks are different, instead of a direct accuracy comparison of our PaI networks with those reported in (Hoang et al., 2023), it is more helpful to compare the drop (or gain) in accuracy from the trained unpruned networks after a certain period of training epochs of the sparse initial Ramanujan networks. This is denoted by $\delta_{acc}$. In terms of the drop in accuracy ($\delta_{acc}$) as compared with the unpruned network, we can see that our method remains close to the accuracy of the original VGG architectures even at $1.7\%$ sparsity, whereas it performs at par and or sometimes even better for the ResNet architectures at low sparsity. The drop in accuracy with respect to a dense network is less or comparable for our proposed method as compared to related techniques for a lower remaining weight ratio.

Table 6: Comparing performance of various PaI methods

| Cifar-10 / CIFAR-VGG (15M) | | | | | VGG-base (135M) | | |
|---|---|---|---|---|---|---|---|
| Reported Results (Hoang et al., 2023) (Network Density $\sim 1.7\%$) | | | | | Our Results (Network Density $\sim 1.7\%$) | | |
| Method | Unpruned | Random | ERK | SNIP | SynFlow | Method | Unpruned | Our method |
| Accuracy | $94\%$ | $83\%$ | $90\%$ | $92\%$ | $92\%$ | Accuracy | $84\%$ | $82\%$ |
| $\delta_{acc}$ | | $11\%$ | $4\%$ | **2%** | **2%** | $\delta_{acc}$ | | **2%** |
| $IMDB$ | | $-0.116971$ | $0.128288$ | $0.163364$ | $0.347729$ | $IMDB$ | | **3.463054** |
| Cifar-10 / ResNet34 / Network Density $= 5.2\%$ | | | | | | | |
| Reported Results (Hoang et al., 2023) (250 epochs) | | | | | Our Results (30 epochs) | | |
| Method | Unpruned | Random | ERK | SNIP | GraSP | Method | Unpruned | Our method |
| Accuracy | $89\%$ | $81\%$ | $86\%$ | $87\%$ | $86\%$ | Accuracy | $83\%$ | $84\%$ |
| $\delta_{acc}$ | | $8\%$ | $3\%$ | $2\%$ | $3\%$ | $\delta_{acc}$ | | **-1% (better)** |
| $IMDB$ | | $-0.018530$ | $0.181492$ | $0.297726$ | $0.188965$ | $IMDB$ | | **3.098840** |
| Cifar-100 / ResNet34 / Network Density $= 5.2\%$ | | | | | | | |
| Reported Results (Hoang et al., 2023) (250 epochs) | | | | | Our Results (30 epochs) | | |
| Method | Unpruned | Random | ERK | SNIP | GraSP | Method | Unpruned | Our method |
| Accuracy | $62\%$ | $52\%$ | $59\%$ | $60\%$ | $59\%$ | Accuracy | $57\%$ | $56\%$ |
| $\delta_{acc}$ | | $10\%$ | $3\%$ | $2\%$ | $3\%$ | $\delta_{acc}$ | | **1%** |
| $IMDB$ | | $-0.018530$ | $0.101492$ | $0.297726$ | $0.188965$ | $IMDB$ | | **3.098840** |

Next, we also compare the IMDB (iterative mean difference of bound) as used by (Hoang et al., 2023) to compare the constructed Ramanujan graph network with the sparse networks obtained by other PaI techniques reported in Table 6. The value is computed as $IMDB = \sum_{K}(\sqrt{d_{left} - 1} + \sqrt{d_{right} - 1} - \lambda_2)$. Here, $d_{left}$ and $d_{right}$ are the degrees of the left and right side of the bipartite Ramanujan Graph that we have constructed and $\lambda_2$ is the second largest eigenvalue by magnitude. $K$ represents the number of layers that we have pruned in our model. The IMDB values for various methods along with our method is also given in Table 6. It can be seen that a much higher IMDB value is obtained in the proposed method as compared to existing PaI techniques. This shows that deterministic Ramanujan graph construction generates better connected sparse networks.

## 6 CONCLUSION

We present a deterministic method for constructing sparse neural network structures which upon weight initialization can be trained to a high accuracy. The method is based on a Ramanujan graph construction technique using Cayley graphs and Ramanujan coverings. Unlike random graph generation, this always results in a structured, symmetric, and regular sparse network. The method is adapted for masking both the fully connected and convolution layers. Experimental results on popular architectures and datasets demonstrate that close to unpruned network accuracy can be achieved using a very sparse network structure.

In future, we would like to exploit the structured sparsity of the graph adjacency matrices for efficient implementation of the training algorithms. We would also extend the method to other recurrent neural architectures.

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
