# OpenReview forum: "Constructing Sparse Neural Architecture with Deterministic Ramanujan Graphs"
_ICLR.cc/2024/Conference — Submitted to ICLR 2024_

### Official Review · Reviewer_t3Ja · 2023-10-31

**Soundness:** 2 fair
**Presentation:** 2 fair
**Contribution:** 2 fair
**Rating:** 3
**Confidence:** 4

**Summary:**

The paper presents a deterministic method for constructing sparse neural network structures which upon weight initialization can be trained to a high accuracy. The method is based on a Ramanujan graph construction technique. Experimental results show that their methods is able to achieve high sparsity without performances losses.

**Strengths:**

1. The proposed methods achieve higher sparsity compared to SNIP and GraSP.

2. The method works in pre-training phase and is weight free, which are two main advatanges.

**Weaknesses:**

1. The experiment is kind of weak. Apart from SNIP and GraSP., there are other pre-training pruning methods as well. Besides, there is also a series of works focused on during-training pruning, such as SET/RigL. I recommend the author give a comprehensive comparison as well.
2. The performance improved compared to existing methods is limited, which means the proposed method only makes a limited contribution.
3. A critical problem of the methods remains that unstructured sparsity is hard for physical acceleration.

**Questions:**

1. What is the computational complexity of the proposed sparsification method to compute the sparse mask?
2. How the sparse training is implemented in the experiments? Can the authors give a detailed explanation?

---

> ### Author Response · Authors · 2023-11-13
> **Response to Comments.**
>
> Thank you for your thoughtful comments on our article, and we appreciate the opportunity to address the concerns you've raised.
>
> Weakness 1: The experiment is kind of weak. Apart from SNIP and GraSP., there are other pre-training pruning methods as well. Besides, there is also a series of works focused on during-training pruning, such as SET/RigL. I recommend the author give a comprehensive comparison as well.
>
> Response: We chose SNIP and GraSP as they are widely used as baselines, even though more competitive techniques exist. We did not compare with during-training pruning methods as they have a higher computational complexity and are not directly comparable.
>
> Weakness 2: The performance improved compared to existing methods is limited, which means the proposed method only makes a limited contribution.
>
> Response: As seen from Table 9, the accuracy drop for our method is close to that of the existing methods (about 1-3%). However, we could achieve this at a significantly lower remaining weight ratio of 1.7% as compared to 5.2% for the other methods reported in Table 9. Thus, our method could obtain a sparse network with a similar accuracy drop.
>
> Weakness 3: A critical problem of the methods remains that unstructured sparsity is hard for physical acceleration.
>
> Response: The proposed deterministic construction technique using Cayley graphs and bi-regular Ramanujan graphs do achieve structured sparsity due to the underlying symmetry and structure of the generated adjacency matrices. See discussion in pages 3 and 5. Even though we have not conducted studies in compiler and hardware optimizations, we believe that structured sparsity is an inherent advantage of the proposed adjacency matrix construction technique.
>
> Question 1: What is the computational complexity of the proposed sparsification method to compute the sparse mask?
>
> Response: Computational Complexity:
> A $m$ by $n$ bipartite graph connects two network layers with $m$ and $n$ nodes respectively.
> 1. The technique for Convolution layers to generate bi-regular bipartite graphs (of order $m$ by $n$) has complexity $O(mn)$. This complexity is due to the creation of the pruning mask which is of size $m \times n$.
> 2. The LPS technique to generate $p+1$ regular Ramanujan graphs  has complexity$ O(q^5 + p^4) = O(q^5) = O(m^{(5/3)})$. Here $m = n = O(q^3)$. This complexity is due to the creation of the PGL_2 group in which first we need to create the generator matrix and then find the equivalence classes which takes time $O(q^4 * q)$. The solution to the four square problem has complexity $O(p^4)$. Since the number of nodes are much less compared to the total parameters, the complexity is not very high.
>
> Question 2: How the sparse training is implemented in the experiments? Can the authors give a detailed explanation?
>
> Response: We generate an initial sparse structure for the network by the Ramanujan graph construction process. Our goal was to demonstrate the effectiveness of this sparse structure. Hence, no specialized sparse training method is used thereafter. The training parameters used are listed in Table 4. One may further exploit the Ramanujan graph structure by optimizations during sparse training. However, that has not been the goal of the current study. We would be very excited to explore these optimizations in future.
>
> We are committed to addressing your concerns and ensuring that the significance of our contributions is clearly conveyed. If you have further suggestions or specific areas you would like us to elaborate on, please feel free to let us know.

---

### Official Review · Reviewer_qgd3 · 2023-10-31

**Soundness:** 2 fair
**Presentation:** 1 poor
**Contribution:** 1 poor
**Rating:** 5
**Confidence:** 2

**Summary:**

This paper proposes to prune / sparsify fully connected and convolutional layers in deep nets using deterministic Ramanujan graphs,

**Strengths:**

Interesting and theoretically well motivated approach.

**Weaknesses:**

While the conceptual idea on its own is interesting and promising, this paper is lacking deeper theoretical insights (actual novel theoretical contributions) and/or more importantly solid experiments that properly compare the proposed methods with recent baselines.

The paper tries to achieve the latter in Table 9, where the achieved pruning and accuracy is compared with three recent baselines. However, the baselines are evaluated on different variants of the initial neural networks (e.g. different number of parameters, etc.) and hence are not really comparable. Besides, the baselines achieve typically much higher (absolute) accuracy, but as as just said they are not really comparable.

Overall, the fact that Ramanujan graphs were already used for pruning (Hoang et al ICLR 2023) makes this contribution rather iterative (going from random Ramanujan graphs to deterministic ones). I am willing to change my score if authors / other reviewers address my concerns and convince me of the significance of the contribution.

Minor comments:
* Wrong use of \citep vs \citet (e.g."[..] data independent contexts Cheng et al. (2023)" must be (Cheng et al., 2023) and many more such examples)
* Perhaps typo in title "Neural Architecture**s**"?

**Questions:**

/

---

> ### Author Response · Authors · 2023-11-13
> **Response to comments.**
>
> Thank you for your thoughtful comments on our article, and we appreciate the opportunity to address the concerns you've raised.
>
> Novelty of our work:
>
> One of the key innovations in our article is the consideration of actual Ramanujan graphs as a PaI technique. Unlike previous works, including those by [4] and [1], our approach introduces deterministic (bi-)regular Ramanujan graphs enhancing the robustness and effectiveness of the initialization process. [4] considered checking for the spectral bound of $\lambda_2 \leqslant  \sqrt{(d1_{avg}-1)} +  \sqrt{(d2_{avg}-1)}$ as a means to ensure Ramanujan property taking inspiration from the fact that for bi-regular graphs this is indeed the case. Hoang et al [1] made it more robust by treating the problem of irregularity by introducing a parameter called $\Delta_{r-imdb}$ which is defined on regular subgraphs within the irregular expanders and ensures that it doesn't descend into complete randomness (see for instance, pg 4 - after Definition 4 and pg - 5 IMDB). They then proceeded to do a comparison test on several architectures including VGG, ResNet etc for Cifar 10 (see Fig - 1 of [1]).
>
> In our case, we mitigate the problem of irregularity by implementing deterministic regular Ramanujan graphs, thus avoiding the issue raised by Hoang et al. [1]. Our Table 4-5 gives the accuracy of our pruned network architecture on Cifar-10 dataset for various base architectures including ResNet. Besides accuracy, we could also achieve a better IMDB value [1] for our network. See Table 6 of revised manuscript.
>
> Experimental comparison:
>
> In our presentation, we intentionally chose to showcase the effectiveness of our method on the VGG architecture by using the VGG-base with 135M parameters. This choice was made to emphasize our ability to implement the method without additional information on dropout and pretraining, which the smaller VGG (15M parameters) architecture utilizes.
>
> Our key observation is that, even with the larger VGG-base, we achieved similar accuracy to the unpruned VGG-base while retaining only 1.7% network density. This highlights the robustness and efficiency of our approach in achieving significant pruning without compromising performance.
>
> For ResNet34, we used the same architecture as Hoang's to facilitate a direct comparison. Notably, after just 30 epochs (compared to 250 in Hoang's article), our model reaches a comparable accuracy to the unpruned network. This result underscores the efficiency of our method in achieving competitive performance with a reduced training time.
> We acknowledge your concern about the comparability of baselines and will ensure that this aspect is appropriately emphasized in the revised version of the paper.
>
> Key Contributions:
>
> The transition from random Ramanujan graphs to deterministic ones is a crucial advancement with significant implications for the properties of network structures. The previous constructions of random Ramanujan networks, while exhibiting certain properties, may lack essential characteristics such as path connectivity and are irregular, even descending into complete randomness, as observed in Hoang et al., necessitating their introduction of the Iterative Mean Difference of Bound (IMDB).  In fact, there is no proper definition of irregular Ramanujan graphs when the minimal degree is small. Both [4] and [1] mention Hoory's work on the bound of the spectral radius of the universal cover $\tilde{G}$ of the irregular graph $G$, but it needs to be added that the theorem only works where the minimal degree is at least 2, see Hoory [3], corollary 5. This is not ensured in the previous works.
>
> In contrast, our proposal of deterministic Ramanujan graph architecture ensures path connectivity, eliminates irregularity concerns, and maintains a high degree of symmetry. This deterministic approach addresses limitations present in random Ramanujan networks, providing a more controlled and predictable foundation for network initialization.
>
> Further, the implementation is the first mathematically rigorous implementation of Ramanujan graph network as the previous approaches couldn't ensure the following at the same time:
> 1. Minimal degree of the created graph networks is at least $2$ and
> 2. The universal cover graphs have spectral radius satisfying Hoory's bound.
> We are committed to addressing your concerns and ensuring that the significance of our contributions is clearly conveyed. If you have further suggestions or specific areas you would like us to elaborate on, please feel free to let us know.
>
> [1] Hoang, Duc NM, et al. "Revisiting pruning at initialization through the lens of Ramanujan graph." ICLR 2023.
>
> [3] Hoory. "A lower bound on the spectral radius of the universal cover of a graph." J. Combinatorial Theory, 2005.
>
> [4] Pal, et al. "A Study on the Ramanujan Graph Property of Winning Lottery Tickets." ICML 2022.

---

> > ### Comment · Reviewer_qgd3 · 2023-11-20
> >
> > Thanks for the reply. Note that this is not my main research field. Having read the many concerns raised by the other reviewers, I trust their judgement. Hence, I keep my score.

---

### Official Review · Reviewer_FU2a · 2023-11-01

**Soundness:** 2 fair
**Presentation:** 2 fair
**Contribution:** 2 fair
**Rating:** 3
**Confidence:** 2

**Summary:**

The work empirically studies a deterministic method for constructing sparse networks. The method is based on some known regular and bi-regular Ramanujan graph construction techniques.  Experiments on Cifar10/Cifar100 show the effectiveness of the proposed method compared to baselines.

**Strengths:**

+ The Ramanujan graph construction techniques are based on recent improvements in graph theory on Ramanujan graphs.
+ Empirical results demonstrate the effectiveness of the proposed method.

**Weaknesses:**

- The introduction of the Ramanujan graph construction techniques is too dense and without a good preliminary, which is hard for ML communities to appreciate in general. Actually, the paper, even for its main text has one paper left to fill in. The paper was written in rush. More details on the adopted techniques should be introduced.

- It confuses people when constructing fully connected layers, one adopts Cayley graphs, while to construct convolution layers, one adopts the approach in Sec 4.3.

- The construction seems not to be able to construct layers with arbitrary sizes.

- Empirical evaluation is not based on larger datasets, such as ImageNet. As I am not an expert who works on network pruning, I am not sure if a larger dataset should be used.

- In section 4.4, q should be larger than l. However, in table 9, q is smaller than l.

**Questions:**

1. Can the authors explain why "In section 4.4, q should be larger than l. However, in table 9, q is smaller than l" ?

2. Can the authors explain why "when constructing fully connected layers, one adopts Cayley graphs, while to construct convolution layers, one adopts the approach in Sec 4.3"?

---

> ### Author Response · Authors · 2023-11-13
> **Response to the questions.**
>
> Thank you for your valuable feedback. We appreciate your comments and would like to clarify the rationale behind our choices.
>
> Weakness 1: The introduction of the Ramanujan graph construction techniques is too dense and without a good preliminary. More details on the adopted techniques should be introduced.
>
> Response: We will add the preliminaries in the revised version.
>
> Weakness 2: It confuses people when constructing fully connected layers, one adopts Cayley graphs, while to construct convolution layers, one adopts the approach in section 4.3.
>
> Response: In our article, we employ Cayley graphs specifically for balanced bipartite graphs when constructing fully connected layers. The reason behind this is that Cayley graphs are regular (and not bi-regular) which is suitable for modelising balanced bipartite graphs.
> However, when dealing with unbalanced bipartite graphs, we shift our approach to the methodology outlined in section 4.3. For unbalanced cases this decision is driven by the need to modelize these graphs using bi-regular structures.
>
> Weakness 3: The construction seems not to be able to construct layers with arbitrary sizes.
>
> Response: Thank you for highlighting this concern. We want to clarify that the ability to construct layers with arbitrary sizes is addressed in our article, specifically in section 4.5. In this section, we point to a polynomial time algorithm that enables the construction of layers with arbitrary sizes and degrees, including bi-regular degrees.
>
> One of the novelties of the article lies in demonstrating how deterministic Ramanujan graphs can be used as PaI techniques without prior knowledge of weights and or dropouts. For instance, a lot of PaI techniques  actually have prior knowledge of dropouts of the base model and start from smaller architectures before employing the respective pruning algorithms. In this case, we don't need it. Of course, if we have smaller architectures, then also our method works.
>
> Weakness 4: Empirical evaluation is not based on larger datasets, such as ImageNet. As I am not an expert who works on network pruning, I am not sure if a larger dataset should be used.
>
> Response: Thank you for bringing up this point. In the context of network pruning, it's common practice not to use larger datasets like ImageNet for empirical evaluation. In general pruning techniques typically focus on optimizing model efficiency and reducing redundancy in parameters, rather than aiming for high accuracy on large-scale datasets. The choice of dataset for evaluation is often influenced by the specific goals of the pruning algorithm. Smaller datasets are frequently used to assess the effectiveness of pruning in terms of parameter reduction and computational efficiency. Our decision to use datasets that align with these objectives is in line with standard practices in the field.
>
> Weakness 5: In section 4.4, q should be larger than l. However, in table 9, q is smaller than l.
>
> Response: Thank you for pointing out this discrepancy. It's important to note that in section 4.4, the condition $q \geq l$ is not a necessary requirement for implementing the technique outlined in section 4.3. The reason behind this flexibility lies in the specific properties of the unbalanced bipartite graph $B$, which has dimensions $q^2$ by $lq$. The critical insight is that if the unbalanced bipartite graph $B$ is Ramanujan, then its transpose, denoted as $B^T$ (with dimensions $lq$ by $q^2$), is also Ramanujan. Therefore, the conditions in Table 9 do not strictly adhere to $q$ being larger than $l$, as the Ramanujan property holds even when $q$ is smaller than  $l$.  We will ensure that this clarification is appropriately reflected in the revised version of the article.
>
> We are committed to addressing your concerns and ensuring that the significance of our contributions is clearly conveyed. If you have further suggestions or specific areas you would like us to elaborate on, please feel free to let us know.

---

### Official Review · Reviewer_Qqf9 · 2023-11-10

**Soundness:** 2 fair
**Presentation:** 2 fair
**Contribution:** 2 fair
**Rating:** 5
**Confidence:** 3

**Summary:**

This paper proposes to use deterministic Ramanujan graphs as a PaI technique for neural architecture sparsification, which can be applied to fully connected and convolutional layers. The proposed method is evaluated on classical architectures for vision tasks.

**Strengths:**

- The proposed technique is theoretically supported by spectral graph theory.
- The adopted deterministic Ramanujan diagram has desirable properties for sparsification.

**Weaknesses:**

- Ramanujan graphs have been well-studied for pruning at initialization [1] and sparse architectures [2].
- The experiments are not well designed, especially for the comparison with other PaI methods and the introduction of metric $\delta_{acc}$.

[1] Hoang, Duc NM, et al. "Revisiting pruning at initialization through the lens of Ramanujan graph." ICLR 2023.
[2] Vooturi, Dharma Teja, Girish Varma, and Kishore Kothapalli. "Ramanujan bipartite graph products for efficient block sparse neural networks." Concurrency and Computation: Practice and Experience 35.14 (2023): e6363.

**Questions:**

- It is very interesting to see that the conclusion here is different from [1], as it claims “not only the Ramanujan property for sparse networks shows no significant relationship to PaI’s relative performance, but maximizing it can also lead to the formation of pseudo-random graphs with no structural meanings”. Can the author elaborate how the property of “deterministic” may mitigate this?
- It seems that Table 9 is not a fair comparison. PaI baselines are applied to CIFAR-VGG, while the proposed method is applied to VGG-base instead, which is 9 times larger than CIFAR-VGG. Meanwhile, the setting adopted for them is also different. The introduced metric $\delta_{acc}$ is inappropriate to evaluate them as the setting and difficulty of pruning are all different.

---

> ### Author Response · Authors · 2023-11-13
> **We highlight the significance of our contribution.**
>
> Thank you for your thoughtful comments on our article, and we appreciate the opportunity to address the concerns you've raised. We want to provide a detailed explanation of our approach and its distinctions from previous methods using Ramanujan graphs for PaI.
>
> Prior approaches to using Ramanujan expander graphs for PaI have predominantly relied on constructions based on random or iterated magnitude pruning techniques. Unfortunately, this approach has led to the formation of irregular graph networks that do not strictly adhere to the rigorous definition of Ramanujan graphs.
>
> It is to be emphasized that a $b$-regular (respectively ($b$,$l$) bi-regular)  bipartite graph is considered Ramanujan if the second largest eigenvalue of its adjacency matrix is less than or equal to $2\sqrt{(b-1)}$ (respectively  $\sqrt{(b-1)} +  \sqrt{(l-1)}$) where $b,l \geq 2$. Irregular Ramanujan graphs lack a singular definition, but a widely acknowledged one, as introduced by Hoory [3], relates the second largest eigenvalue of an irregular bipartite graph to the spectral radius of its universal cover graph. This method relies on the condition that the minimal degrees of the obtained graphs need to be larger than 2. See Corollary 5. This has been used in both [4] and [1]. It can only be applied when the minimal degrees of the obtained graphs are larger than 2.
>
> Hoang et al [1] mitigates the issue of irregularity by considering regular subgraphs (of regularity $\geq 3$) of the irregular structures so formed and then using their IMDB which is essentially computing the sum $\delta_r$ for these regular subgraphs. We take a different, approach by implementing a deterministic regular Ramanujan graph architecture for PaI. This is not contradictory to [1]; rather, it complements it. In our case, our initial graphs are inherently and rigorously Ramanujan, avoiding the complications associated with irregularity.
>
> Further, it should be emphasized that the existence of these Ramanujan graphs are not purely random. Random regular graphs are not known to be Ramanujan (they are weakly Ramanujan viz. Freidman's proof of Alon's conjecture [5]), and their construction poses challenges. Our method provides an explicit implementation of networks modeled by deterministic Ramanujan graphs, addressing the difficulties associated with randomness in previous approaches.
>
> Response to Weaknesses:
> Table 9 reports the accuracy of the dense as well as sparse networks. The metric $\delta_{acc}$ is additionally presented to represent the drop in classification accuracy. We have compared with the well known PaI methods namely, random, ERK, SNIP, and GraSP.
>
> Response to Questions:
> Our paper complements the observations of [1] rather than contradicting it. The conclusion of [1] is not that Ramanujan property has no correlation with performance. Refer to sec 4.1, Figure 1 and Figure 3 of [1] for the relevant results in [1]. It is pointed out that characterization of Ramanujan property of irregular graphs in terms of the bound of the largest non-trivial eigenvalue $\Delta_r$ produces pseudo-randomness in the generated networks. The measures IMDB and NaRC, proposed in [1] are found to have a better correlation with network performance. This motivates us to avoid such pseudo-random generations and use deterministic regular Ramanujan graph based constructions. See Table 6.
>
> We appreciate your scrutiny of Table 9 and understand your concern about the comparability of the baselines. We intentionally chose to showcase the effectiveness of our method on the VGG architecture by using the VGG-base with 135M parameters. This choice was made to emphasize our ability to implement the method without additional information on dropout and pretraining, which the smaller VGG (15M parameters) utilizes.
>
> Our key observation is that, even with the larger VGG-base, we achieved similar accuracy to the unpruned VGG-base while retaining only 1.7% network density. For ResNet34, we used the same architecture as [1] to facilitate a direct comparison. Notably, after just 30 epochs (compared to 250 in [1]), our model reaches a comparable accuracy to the unpruned network. This result underscores the efficiency of our method in achieving competitive performance with a reduced training time. We acknowledge your concern about the comparability of baselines and will ensure that this aspect is appropriately emphasized in the revised version of the paper.
>
> We are committed to addressing your concerns and ensuring that the significance of our contributions is clearly conveyed. If you have further suggestions or specific areas you would like us to elaborate on, please feel free to let us know.
>
> [3] Hoory. "A lower bound on the spectral radius of the universal cover of a graph." J. Combinatorial Theory, 2005.
>
> [4] Pal, et al. "A Study on the Ramanujan Graph Property of Winning Lottery Tickets." ICML 2022.
>
> [5] Friedman. “Relative expanders or weakly relatively Ramanujan graphs.” Duke Math J. 2003.

---

> > ### Comment · Reviewer_Qqf9 · 2023-11-23
> > **Followup on Author Response**
> >
> > Thank the authors for providing clarification and addressing my concerns. I believe this work should be recognized for employing the deterministic regular Ramanujan graph for PaI, which aligns with the findings in [1]. My only remaining concern is with the experiment design, where the authors may need to put more effort into showing the benefits of the proposed methods in fair comparisons and in different settings/tasks. I raised my rating to 5 accordingly.

---

### Meta-Review · Area_Chair_pyKn · 2023-12-02

**Metareview:**

This paper introduces deterministic Ramanujan graphs as a Pruning and Initialization (PaI) technique for neural architecture sparsification, applicable to both fully connected and convolutional layers. The method leverages established regular and bi-regular Ramanujan graph construction techniques. Through empirical evaluation on traditional vision task architectures, the proposed approach demonstrates its efficacy on datasets such as Cifar10/Cifar100 when compared to baseline methods.

While the concept of employing Ramanujan graphs is intriguing, the paper faces challenges regarding novelty, particularly due to a parallel work presented at ICLR 2023. Additionally, the reviewers highlight that the experimental section requires substantial enhancements. Therefore, a significant revision is recommended. Authors are encouraged to address the reviewers' feedback comprehensively, refining the experimental aspects, and consider resubmitting the paper to a forthcoming venue.

**Justification For Why Not Higher Score:**

All reviewers put either reject or weak reject, and I support their review.

**Justification For Why Not Lower Score:**

N/A

---

### Decision · Program_Chairs · 2024-01-16

Reject